# Cocaine Destroys Gray Matter Brain Cells and Accelerates Brain Aging

**DOI:** 10.3390/biology12050752

**Published:** 2023-05-21

**Authors:** Iman Beheshti

**Affiliations:** 1Department of Human Anatomy and Cell Science, Rady Faculty of Health Sciences, University of Manitoba, Winnipeg, MB R3E 0J9, Canada; Iman.beheshti@umanitoba.ca; 2Neuroscience Research Program, Kleysen Institute for Advanced Medicine, Health Sciences Centre, Winnipeg, MB R3E 3J7, Canada

**Keywords:** anatomical MRI, gray matter changes, brain age estimation, cocaine, voxel-based morphometry

## Abstract

**Simple Summary:**

The consumption of cocaine is linked to a range of detrimental outcomes, including addiction, cognitive deficits, and an elevated risk of developing psychiatric disorders. This study sought to assess the effect of cocaine on brain structure by utilizing both statistical (e.g., voxel-based morphometry) and machine learning (e.g., brain age estimation) models. The results from our experiment showed that individuals with cocaine use disorder (CUD) had a significantly reduced amount of gray matter (GM) when compared to age- and sex-matched healthy controls (HCs), implying GM deterioration. In addition, the machine learning model revealed that CUD patients had a higher brain age than that of the HCs, indicating accelerated aging. By shedding light on the adverse effects of cocaine on brain anatomy and the aging process, this study greatly contributes to our understanding of the neurological mechanisms underlying CUD, thus expanding our knowledge in this area.

**Abstract:**

**Introduction:** Cocaine use disorder (CUD) is a substance use disorder characterized by a strong desire to obtain, consume, and misuse cocaine. Little is known about how cocaine affects the structure of the brain. In this study, we first investigated the anatomical brain changes in individuals with CUD compared to their matched healthy controls, and then explored whether these anatomical brain abnormalities contribute to considerably accelerated brain aging among this population. **Methods**: At the first stage, we used anatomical magnetic resonance imaging (MRI) data, voxel-based morphometry (VBM), and deformation-based morphometry techniques to uncover the morphological and macroscopic anatomical brain changes in 74 CUD patients compared to 62 age- and sex-matched healthy controls (HCs) obtained from the SUDMEX CONN dataset, the Mexican MRI dataset of patients with CUD. Then, we computed brain-predicted age difference (i.e., brain-PAD: the brain-predicted age minus the actual age) in CUD and HC groups using a robust brain age estimation framework. Using a multiple regression analysis, we also investigated the regional gray matter (GM) and white matter (WM) changes associated with the brain-PAD. **Results**: Using a whole-brain VBM analysis, we observed widespread gray matter atrophy in CUD patients located in the temporal lobe, frontal lobe, insula, middle frontal gyrus, superior frontal gyrus, rectal gyrus, and limbic lobe regions compared to the HCs. In contrast, we did not observe any swelling in the GM, changes in the WM, or local brain tissue atrophy or expansion between the CUD and HC groups. Furthermore, we found a significantly higher brain-PAD in CUD patients compared to matched HCs (mean difference = 2.62 years, Cohen’s d = 0.54; *t*-test = 3.16, *p* = 0.002). The regression analysis showed significant negative changes in GM volume associated with brain-PAD in the CUD group, particularly in the limbic lobe, subcallosal gyrus, cingulate gyrus, and anterior cingulate regions. **Discussion**: The results of our investigation reveal that chronic cocaine use is linked to significant changes in gray matter, which hasten the process of structural brain aging in individuals who use the drug. These findings offer valuable insights into the impact of cocaine on the composition of the brain.

## 1. Introduction

Cocaine, a potent stimulant drug, is primarily sourced from the coca plant that is grown in South America. Dating back to ancient times, it has a lengthy history of use for both medicinal and recreational purposes. Nevertheless, the widespread consumption of cocaine has become a significant matter of concern for public health due to its addictive attributes and the profound adverse consequences it inflicts on individuals, families, and communities. It has been estimated that 18 million people worldwide are currently using cocaine, with the majority of users located in Latin America, North America, and Europe, according to the World Drug Report 2021 published by the United Nations Office on Drugs and Crime (UNODC; https://www.unodc.org/unodc/data-and-analysis/wdr2021.html, accessed on 15 April 2023). Cocaine usage has a profound and broad impact on society. Cocaine users are at a heightened risk of encountering a wide range of health issues, including cardiovascular disease, respiratory failure, and mental health disorders such as anxiety and depression. In addition, the economic ramifications of cocaine consumption are considerable, as it results in elevated healthcare costs, criminal behavior, and lost productivity. Cocaine is one of the most powerful stimulants that directly affect the brain, elevate a person’s mood, and create great euphoria. Cocaine usage causes blood vessels to constrict, as well as an increase in body temperature, heart rate, blood pressure, and metabolism. Cocaine raises the amount of dopamine and prevents its reabsorption in the brain, disrupting the function of nerve receptors in the long term [1]. The vast majority of individuals who use cocaine do not exhibit a substance use disorder. However, there is an increased risk of dependence with heavier use that can lead to major health issues that affect mood, emotions, cognition, mental health, behavior, and physical health [2,3]. Cocaine use disorder (CUD) is a condition in which an individual repeatedly uses cocaine despite its negative consequences on physical and mental health. As per the DSM-5, CUD is categorized as a substance use disorder and is acknowledged as a psychiatric diagnosis. CUD is associated with cognitive abnormalities in verbal memory and executive functioning during withdrawal that may be recoverable depending on usage frequency and the length of abstinence [4,5,6]. Cocaine can adversely affect the brain through various biological mechanisms, such as oxidative stress, inflammation, and neurotoxicity. Oxidative stress occurs due to an imbalance between the formation of reactive oxygen species (ROS) and the body’s ability to eliminate them [7]. Cocaine increases the production of ROS, which in turn leads to oxidative damage to brain cells [8]. Moreover, cocaine usage can induce an inflammatory response within the brain [9], resulting in a multitude of negative consequences, including damage to brain cells. Neurotoxicity is another direct consequence of cocaine use [10,11], as it stimulates the release of neurotransmitters such as dopamine, serotonin, and norepinephrine, which can cause excitotoxicity, leading to harm or demise of brain cells.

Neuroimaging tools have been proposed as a robust technique that allows us to explore drug actions and repercussions as they act and persist in the brain [12,13]. In the context of CUD, several studies have investigated the effects of cocaine on the brain using various brain imaging modalities [14,15,16,17,18]. For example, Ersche et al. [17] reported a decreased gray matter (GM) volume in the cerebellar cortex, orbitofrontal, insular, cingulate, and temporoparietal areas, as well as an increased GM volume in the basal ganglia area in chronic cocaine users compared to controls. A recent inquiry was conducted to explore the differences in diffusion tensor imaging (DTI) metrics between those who suffer from chronic cocaine use and those who are classified as healthy controls [19]. The findings revealed that chronic cocaine users displayed significantly reduced signals of fractional anisotropy (FA) and axial diffusivity (AD) in several regions, such as the right inferior, splenium of the corpus callosum, body, anterior, posterior, and superior corona radiata regions, in contrast to the healthy control cohort [19]. Compared to non-users, cocaine users have exhibited a reduction in cortical thickness in the lateral frontal regions and a decrease in cortical surface area in the anterior cingulate cortex [20]. Another study [16] stated that cocaine abuse is associated with a significant decrease in FA signals in the inferior frontal white matter, as well as trends towards reduced GM and white matter (WM) volumes in the same brain area.

Despite these findings, we still know very little about the long-term effects of cocaine on the anatomy of the brain and its mechanisms. Anatomical magnetic resonance imaging (MRI) investigations, along with advanced MRI analysis techniques, could enhance our understanding of the impact of CUD on brain structure and the pathophysiology of CUD. This could lead to better treatment approaches. In addition, combining neuroimaging methods with machine learning algorithms has successfully offered adequate techniques for assessing brain health [21]. One of these techniques is brain age estimation, or the “brain age” biomarker, which provides informative information about the rate of brain aging and global brain health, not only among healthy subjects but also among patients with different neurological and non-neurological disorders [22]. It has been demonstrated that structural brain abnormalities are associated with faster brain aging in various neurological disorders, while certain practices such as meditation can significantly slow down this process [22]. To date, the brain age estimation paradigm has been widely used to assess the impact of different neurological diseases—e.g., Alzheimer’s disease [23,24,25], epilepsy [26], Parkinson’s disease [27,28] and schizophrenia [29,30]—on brain health status. 

In this work, we aimed first to explore and validate the effects of CUD on brain volume and microstructure using advanced brain imaging techniques. To this end, we used whole-brain voxel-based morphometry (VBM) and deformation-based morphometry (DBM) analyses to identify GM/WM and macroscopic anatomical changes in a group of CUD patients (*n* = 74) competed to age- and sex-matched healthy controls (HCs) (*n* = 62), respectively. At the second level, we used the brain age estimation paradigm to quantify the impact of cocaine usage on global brain health and the degree of brain aging in individuals with CUD. In addition, another goal of this study was to investigate the brain morphological changes associated with brain-PAD (i.e., the brain-predicted age minus the actual age) in CUD and matched HCs. Based on the relevant literature, we hypothesized that cocaine consumption has a significant negative impact on brain structure, and individuals with CUD experience accelerated brain aging compared to their matched healthy controls.

## 2. Materials and Methods

### 2.1. Participants and MRI Acquisition

We recruited 74 CUD participants and 64 HCs with available anatomical MRI scans from the SUDMEX CONN dataset, the Mexican MRI dataset of patients with CUD. Two HCs were excluded from this study due to missing age and sex data. The two groups were matched in terms of age, sex, and handedness. In the CUD group, the utilization rate was a minimum of three times a week, with a maximum of 60 consecutive days of abstention within the previous year. The CUD participants had not consumed any drugs before or on the day of the study and they had not partaken in the consumption of any drugs during the time of acquisition. More information about the characteristics of the individuals who participated in this study is available in [31].

Anatomical MRI scans were obtained from a 3T scanner (3D FFE SENSE sequence, repetition time = 7 ms, echo time = 3.5 ms, voxel size of 1 mm × 1 mm × 1 mm, flip angle = 10°, matrix = 24 cm × 24 cm, field of view = 24 cm × 24 cm, number of slices = 180, gap = 0, and scan time = 3.19 min) manufactured by Philips (Philips Healthcare, Best, Netherlands & Boston, MA, USA). The MRI scans of five participants were acquired with a voxel size of 0.75 mm × 0.75 mm × 1 mm. More information about the SUDMEX CONN dataset is provided in [31] and OpenNeuro (https://openneuro.org/datasets/ds003346/, accessed on 10 December 2022). Table 1 summarizes the demographics and characteristics of patients with CUD and their respective HCs used in this study.

### 2.2. MRI Preprocessing

The anatomical MRI scans were processed using the CAT12 toolbox (http://www.neuro.uni-jena.de/cat/, accessed on 15 December 2022), which is an extension of the Statistical Parametric Mapping (SPM12) software package (https://www.fil.ion.ucl.ac.uk/spm/software/spm12/, accessed on 15 December 2022). All anatomical MRI scans were preprocessed using the following steps: bias correction, MRI segmentation into GM, WM, and cerebrospinal fluid (CSF) images, DARTEL normalization to MNI space (voxel size = 1.5 mm × 1.5 mm × 1.5 mm), and modulation. We also generated the Jacobian determinant (JD) images. Besides the visual evaluation, the quality of the images was evaluated using the “Check Homogeneity” feature within the CAT12 toolbox. This function was developed to detect MRI images that may contain artifacts or other forms of variability in voxel intensity values, which could potentially compromise the accuracy and consistency of subsequent analyses. All the generated images were smoothed with an 8 mm full-width-half-maximum Gaussian smoothing kernel. Total intracranial volume (TIV) was also computed by CAT12. The GM and WM density images were used for VBM analysis, and the JD images were used for DBM analysis.

### 2.3. Brain Age Calculation

To build our brain age estimation model, we used anatomical MRI scans of 876 cognitively healthy subjects obtained from the Open Access Series of Imaging Studies (OASIS) (https://www.oasis-brains.org/, accessed on 10 December 2022) and the IXI (http://brain-development.org/ixi-dataset/, accessed on 10 December 2022) dataset. We randomly divided these data into training (90% of the data, *n* = 789, mean age = 47.61± 19.47 years) and validation sets (10% of the data, *n* = 87, mean age = 46.16 ± 19.75 years). For the prediction of brain age, we utilized the smoothed images of GM, WM, and CSF that were resampled to an isotropic spatial resolution of 8 mm as features. In addition to brain features, sex, TIV, scanner vendor, and field strength were also considered in the prediction model. To predict brain age, we used a support vector regression (SVR) algorithm, as it has been widely used in neuroimaging-driven studies for estimating brain age [32]. The SVR algorithm was executed through the utilization of MATLAB software, specifically the “*fitrsvm*” function with a linear kernel and automatic KernelScale. A validated bias adjustment technique, as described in [33], was also applied to the model to compute the bias-free brain age values. The prediction accuracy in the training and hold-out sets was quantified based on the mean absolute error (MAE) and the coefficient of determination (R^2^) between the model-estimated age and the chronological age. Of note, the accuracy of the prediction in the training set was evaluated through the implementation of a 10-fold cross-validation approach. We computed the brain-PAD (i.e., chronological age subtracted from model-estimated age) in each group based on the mean and 95% confidence interval (CI). 

### 2.4. Statistical Analysis

After preprocessing the MRI data, the GM, WM, and JD images underwent independent t-test analysis in SPM12 to identify morphological and anatomical differences between the HC and CUD groups. We adjusted the peak-level *p*-value thresholds at <0.001 (uncorrected) and considered the clusters with q < 0.05 (cluster-level FDR-corrected) as significant. A voxel-based analysis of the entire brain was performed in order to identify regional changes in the GM, WM, and JD images.

Sex, the subject’s age, and TIV were considered as covariates in our VBM analysis, whereas sex and the subject’s age were taken into account in the DBM analysis. An independent student *t*-test and Cohen’s *d* were used to compute the mean difference and effect size between HC and CUD groups in terms of brain-PAD, respectively. The statistical analyses were conducted in MATLAB with a significant level of *p* < 0.05. In addition, we used multiple regression analysis in SPM12 with a family-wise error (FWE) threshold of *p* < 0.05 to investigate the alterations in GM and WM associated with brain-PAD in HC and CUD groups, respectively. In the regression analysis, the subject’s age, sex, and TIV were considered in the matrix design and the extent threshold was set at 50 voxels.

## 3. Results

### 3.1. VBM and DBM Analysis

Compared with the HCs, the CUD patients showed a significant GM reduction in the temporal lobe, frontal lobe, insula, middle frontal gyrus, superior frontal gyrus, rectal gyrus and limbic lobe regions. Figure 1 and Table 2 show substantial GM atrophies detected by our whole-brain VBM analysis in the CUD patients versus HCs. The opposite contrast showed no significant GM swelling in the CUD patients when compared to the HCs. In the CUD group compared with the HCs or in the reverse contrast, the VBM and DBM analyses showed no significant alterations in WM and deep brain structures, respectively. 

### 3.2. Brain Age Values

Our brain age estimation model accurately predicted chronological age in the training dataset through the 10-fold cross-validation strategy (MAE = 4.53 years, R^2^ = 0.91, with a mean brain-PAD of 0.00 [95% CI −0.45, 0.45] years). The model produced a comparable output on the validation set (MAE = 4.39 years, R^2^ = 0.92, with a mean brain-PAD of −0.29 [95% CI −1.52, 0.93] years). This model was used to predict brain age in CUD patients and matched HCs. The brain-PAD of CUD participants was significantly higher than that of matched HCs (mean difference = 2.62 years, Cohen’s d = 0.54; *t*-test (134) = 3.16, *p* = 0.002). The mean brain-PAD in the CUD group was 2.61 years (95% CI 1.53, 3.69), while it was 0.001 years (95% CI −1.12, 1.26) in the matched HC group. There was an insignificant difference between males and females in the CUD (*t*-test (72) = 1.10, *p* = 0.27) and matched HC groups (*t*-test (60) = 0.98, *p* = 0.32) in terms of brain-PAD. Figure 2 shows the grouped data plots displaying the brain-PAD values for CUD and HC participants. 

### 3.3. Regional Relationship between GM and WM Alterations with Brain-PAD

In order to examine the alterations in GM and WM that are linked to brain-PAD, a multiple regression analysis was conducted separately on both the HC and CUD groups. Significant negative changes in GM were observed in the limbic lobe, subcallosal gyrus, cingulate gyrus, and anterior cingulate regions of individuals with CUD in association with brain-PAD. No significant positive interaction effect was found between GM volumes and brain-PAD in individuals with CUD. No significant WM changes were observed to be associated with brain-PAD in CUD, but we observed slight negative WM changes in the HC group. Additionally, no substantial interaction effects between GM volumes and brain-PAD were observed in the HC group. The results of the multiple regression analysis in both the CUD and matched HC groups are presented in Figure 3 and Table 3. 

## 4. Discussion

The objective of this study was to assess the impact of cocaine on brain structure and global brain health. At the first level, we performed VBM and DBM analyses to identify the brain changes in CUD groups in comparison to sex- and age-matched HCs. Of note, the VBM technique measures changes in the volume and density of GM and WM in different brain areas, whereas DBM provides information about changes in the physical shape and deformation of brain structures. When we compared the GM differences between CUD patients and healthy controls, we observed significant morphological changes in the CUD group, including atrophy in several areas such as the temporal lobe, frontal lobe, insula, and superior temporal gyrus (Table 2). These brain areas are mainly associated with processing emotions, language, attention, higher cognitive functions (e.g., working memory), and making decisions. These findings are in agreement with other clinical studies that have reported impairments in emotional recognition [34], language proceeding and cognitive functions (e.g., verbal learning/memory attention, and working memory) in individuals with CUD [35]. The results of our investigation regarding the regions of GM atrophy in CUD are consistent with previous research that has identified significant GM atrophy in cocaine users, particularly in the insula, anterior cingulate cortex, orbitofrontal cortex, and superior temporal cortex regions [36]. Although previous research has shown that chronic cocaine users have an increase in gray matter volume in the basal ganglia region [17], our VBM analysis did not reveal any significant GM enlargement in CUD patients compared to healthy controls. Several explanations are possible for this ambiguity: sample size, the characteristics of patients with CUD, and pre-processing software. For example, it has been shown that even two versions of a processing program can lead to dissimilar results, ultimately influencing the interpretation of findings [37]. Interestingly, the VBM analysis showed no significant alterations in WM in the CUD group compared to the HCs, or in the reverse comparison. This finding could imply that cocaine has a negligible effect on the WM area, which is primarily composed of long-range myelinated axons. Furthermore, DBM analysis did not reveal any significant differences in deep brain structures between the CUD group and the HCs. Of note, DBM is an advanced MRI analysis approach that detects brain atrophy in subcortical regions more sensitively than VBM. These findings would imply that cocaine has a deleterious impact on GM, which consists of neural cell bodies, axon terminals, dendrites, and all nerve synapses. It is important to note that GM loss is associated with a number of aging symptoms, such as memory issues and other deteriorating cognitive capacities. These results are in line with the clinical signs of memory impairment in the CUD The results are in agreement with the clinical symptoms of memory deterioration in individuals suffering from CUD as documented in earlier investigations [4,5,6].

A secondary goal was to assess whether cocaine could accelerate brain aging. To this end, we used the brain age metric that has been widely used to quantify the amount of deceleration or acceleration of brain aging [32]. Brain age estimation is a technique that utilizes supervised machine learning algorithms to analyze brain scans and determine the biological age of an individual’s brain based on their brain features. This technique involves compressing the whole-brain information into a single numerical value, known as the brain-PAD, which represents the difference between an individual’s predicted brain age and their chronological age. A negative brain-PAD score suggests a younger-looking brain, which is advantageous, while a positive score may indicate a divergence between predicted and chronological age, potentially indicating an increased risk of age-related cognitive decline or disease. Brain age estimation has proven useful in detecting early signs of neurodegenerative diseases, assessing overall brain health and aging, predicting cognitive decline, and identifying the impact of lifestyle factors such as diet, exercise, mental health, and stress on brain aging [21]. Further information about the brain age estimation technique and its potential application in clinical settings can be found in [21,22]. 

At the group level, we found that patients with CUD had a positive brain-PAD that was 2.62 years greater than that of matched HCs (Cohen’s d = 0.54; *t*-test (134) = 3.16, *p* = 0.002). Note that a positive brain-PAD value indicates an older-appearing brain [22]. In other words, CUD patients were predicted to have brains that were approximately 2.5 years older than their chronological age based on the brain age model. Consequently, these findings support our hypothesis that individuals with CUD experience considerably faster brain aging. A mean brain-PAD of +2.5 years in CUD patients is similar to that of some neurological diseases, such as Parkinson’s disease (+2.5 years) [27], schizophrenia (+2.56 years) [38], major depressive disorder (+2.78 years) [39], and anxiety (+2.91 years) [39]. Interestingly, we found no significant difference between males and females in the CUD group in terms of brain age values (*t*-test (72) = 1.10, *p* = 0.27), suggesting that both sexes experience the same level of brain deterioration due to CUD. In this study, we employed a standard SVR algorithm for predicting brain age values, as this approach is commonly used in neuroimaging studies for estimating brain age [32]. However, it is crucial to acknowledge that several factors, such as the choice of prediction algorithm, image preprocessing, MRI protocols, and bias adjustments, can influence brain age values and affect the interpretation of the results. 

Furthermore, we conducted multiple regression analyses to reveal the regional changes in GM and WM associated with brain-PAD (Table 3). The regression analysis revealed a negative association between brain-PAD and GM in the limbic lobe, which is associated with cognitive and emotional functions. This finding is consistent with prior research that has identified a correlation between cocaine use and cognitive abilities [6]. In the CUD group, we also observed a negative association between brain-PAD and GM changes in the subcallosal gyrus, that is involved in processing emotions, behavior regulation, depression, and anhedonia (i.e., reduced motivation or interest in activities). This finding is in line with previous studies that investigated depression [40] and anhedonia [41] among cocaine users. Moreover, positive brain-PAD scores in CUD were associated with a decrease in GM in the cingulate gyrus area, which is important in emotional processing and cognition. Damage to the cingulate gyrus might impair the ability to respond to some stimuli, leading to aggressive behavior, decreased emotional expressiveness, or shyness. Indeed, damage to the cingulate gyrus has been linked with increased aggression and impulsivity, as well as decreased empathy and social awareness [42]. These changes may result from an inability to regulate emotional responses or to effectively interpret social cues, which can lead to difficulties in social interactions. Thus, damage to the cingulate gyrus can have significant negative impacts on a person’s emotional and behavioral functioning, underscoring the significance of this brain region in the regulation and processing of emotions [43].

The identification of brain regions that are linked to CUD can significantly contribute to our comprehension of the impact of cocaine on the brain’s mechanisms. By investigating the functional role of these brain regions in the disease progression, as well as the molecular and cellular mechanisms disrupted in these regions, we may gain insight into how CUD develops. This knowledge can also help determine the specific risks, such as cognitive decline, depression, and anxiety, that patients with CUD are exposed to. Additionally, it can aid in the development of innovative treatments, including repetitive transcranial magnetic stimulation (rTMS) and high-definition transcranial direct current stimulation (HD-tDCS) therapies [44]. Of note, identifying the most accurate stimulation target regions is a challenging and complex task in rTMS and HD-tDCS therapies. To the best of our knowledge, this study is the first to quantify the effects of cocaine use on global brain health in terms of brain age metric.

There are some limitations to the current study that should be taken into account. One limitation of our study might be the gender disparity in the subject groups, with males making up the majority of the CUD group. Although we did not observe a statistically significant difference between males and females in the CUD group in terms of brain age values, further research involving an equal representation of both genders is required to confirm this finding. The sample size used in this study for brain age analysis was limited, consisting of only 74 CUD participants and 64 healthy controls. To gain further insight into the effects of cocaine on brain structure, larger sample sizes should be utilized in future research. Additionally, it is important to acknowledge that this study utilized a specific dataset of CUD patients and HCs from Mexico [31]. The race or ethnicity of individuals who use cocaine may have an impact on their behavioral tendencies, and future studies should aim to investigate the effects of cocaine on brain structure in different racial and ethnic groups. 

The CUD group in this study was composed of young patients with a mean age of 30.99 ± 7.25 years. Determining the impact of aging on brain health among older CUD patients is important to investigate in future studies. We acknowledge that the use of a specific imaging protocol and MRI scanner may limit the generalizability of our findings to other imaging protocols or scanners. Therefore, it is crucial to conduct further studies with larger sample sizes obtained from different MRI protocols to validate our results. Moreover, as statistical significance can occasionally be impacted by a number of variables, we should be cautious when interpreting the significance of the results. It is documented that individuals with obesity have older brains, indicating lower brain health, which is likely owing to GM and WM atrophy caused by obesity. However, obesity-related impairments in brain health may be reversed by substantial weight loss [45]. In this study, we demonstrated that cocaine has a negative impact on brain structure, particularly the cortex area. However, it is unclear whether these brain injuries due to cocaine consumption are reversible, for example, by quitting cocaine use and adopting a healthy diet, stress management, and exercise. Further study is required to understand the short- and long-term effects of cocaine therapy on brain structure. We assessed the influence of cocaine on brain structure in this cross-sectional study. Longitudinal investigations are needed to determine the course of GM atrophy in CUD over time. Furthermore, additional research should be conducted to investigate the impact of cocaine on functional brain deficits, such as functional connectivity. In particular, investigating functional connectivity through resting-state fMRI data is essential to comprehending the arrangement and communication of neural networks in the brain, as well as the functional interactions between different brain regions [46]. 

## 5. Conclusions

In this study, we investigated the impact of cocaine on brain structure. We explored the GM and WM volume abnormalities in individuals with CUD compared with age- and sex-matched HCs at the voxel level. We also demonstrated that CUD is associated with accelerated brain aging of approximately +2.5 years, which can result in a wide range of cognitive impairments (such as memory loss and reduced attention span), behavioral issues, and emotional impairments. We identified the brain regions that are associated with brain-PAD in CUD. The identification of the particular brain regions affected by CUD could greatly enhance our comprehension of the disorder’s effects on brain structure and potentially lead to the development of novel therapeutic approaches for individuals with this condition.

## Figures and Tables

**Figure 1 biology-12-00752-f001:**
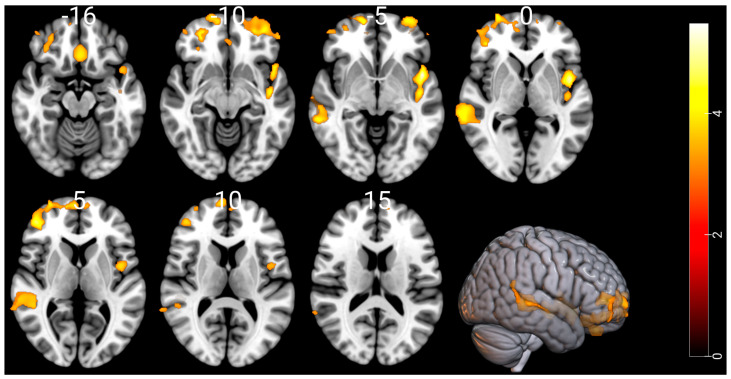
Comparison of gray matter volume by VBM among 74 CUD patients and 62 HCs. Significant gray matter atrophy was observed in the temporal lobe, frontal lobe, insula, middle frontal gyrus, superior frontal gyrus, rectal gyrus, and limbic lobe regions in CUD patients compared to age- and sex-matched HCs. Substantial alterations are displayed as colored brain areas. The T-map was generated based on FDR-uncorrected at *p* < 0.001 with an extent threshold of k > 900. The color bar represents the *t*-test between the two groups. No gray matter swelling was detected in any region.

**Figure 2 biology-12-00752-f002:**
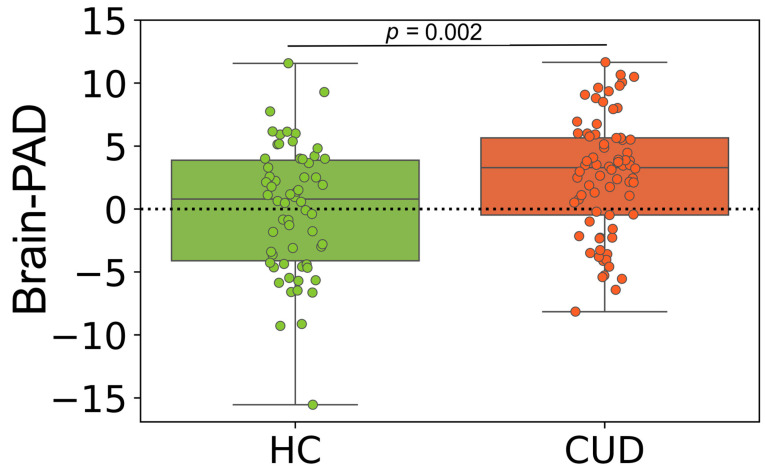
Box plots displaying the grouped brain-PAD values (years) in individuals with CUD (*n* = 74) and matched HCs (*n* = 62). The reference line is represented by the black dashed line (*y* = 0). The Student t-test was used to conduct the statistical test between groups.

**Figure 3 biology-12-00752-f003:**
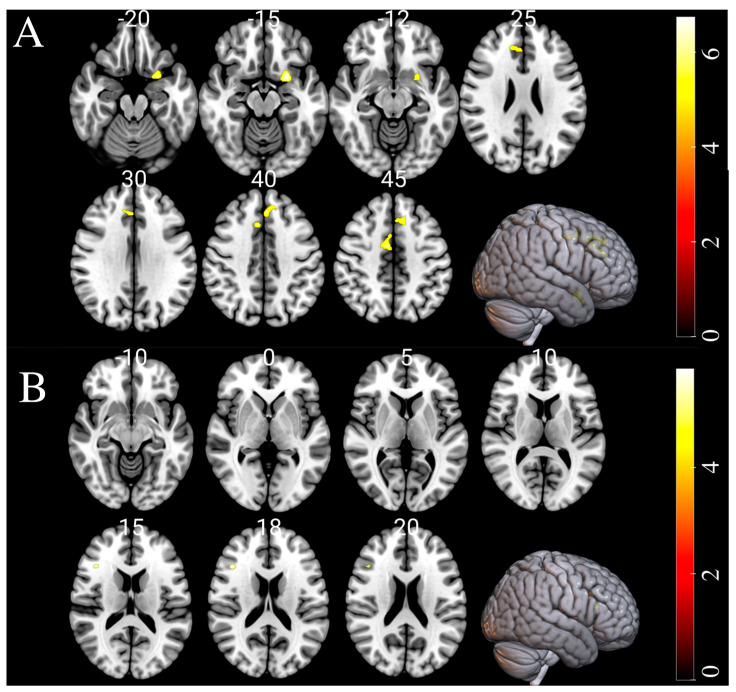
The results of negative gray matter (GM) and white matter (WM) volumes alterations associated with brain-PAD in CUD and HC groups, respectively. Substantial alterations are displayed as colored brain areas. These maps were generated using multiple regression analysis in SPM12 with a family-wise error (FWE) threshold of *p* < 0.05 and an extent threshold of K = 50. (**A**) GM in CUD and (**B**) WM in HC.

**Table 1 biology-12-00752-t001:** The demographics of HC and CUD participants used in this study.

	HC	CUD	*p*
Number	62, 83% male	74, 88% male	
Age, years	30.60 ± 8.26	30.99 ± 7.25	0.77
Education, years	12.73 ± 3.34	11.26 ± 3.17	0.01
Years of consumption	n.a	10.07 ± 6.79 ^a^	n.a
Cocaine age onset	n.a	20.91 ± 5.59 ^a^	n.a
Weekly dose	n.a	2.97 ± 1.21 ^b^	n.a

HC, healthy control; CUD, cocaine use disorder; *p*, *t*-test between HC and CUD; n.a, not available. ^a^ Data missing in 4 participants, ^b^ Data missing in 5 participants.

**Table 2 biology-12-00752-t002:** Clusters of gray matter atrophies detected by the VBM analysis using SPM12 software (74 CUD patients vs. 62 matched HCs).

Cluster	Region	BA	Cluster Size (No. of Voxels)	q (FDR)	Hemisphere	MNI Coordinates (x, y, z)	T Value (Peak Voxel)
1	Temporal Lobe/Insula/Superior Temporal Gyrus/Superior temporal gyrus/Temporal pole	13/22/38	1579	0.005	L	−45, 4, −3	5.58
					L	−44, −10, −8	4.14
					L	−44, 12, −16	3.66
2	Frontal_Mid_2/Middle Frontal Gyrus/Superior Frontal Gyrus	10/11	2558	0.001	R	45, 45, 4	4.58
					R	15, 62, −4	4.59
					R	30, 66, 0	4.45
3	Temporal Lobe/Temporal_Mid/Middle Temporal Gyrus/Superior Temporal Gyrus	21/22	1726	0.005	R	66, −34, 0	4.41
					R	60, −39, −4	4.41
					R	60, −27, −2	4.06
4	Frontal Lobe/Middle Frontal Gyrus/Frontal_Mid_2/Superior Frontal Gyrus	10/11	986	0.028	L	−36, 51, −9	4.28
					L	−27, 51, −9	4.23
					L	−30, 63, −6	4.20
5	Frontal Lobe/Medial Frontal Gyrus/Rectal Gyrus/Limbic lobe	11/25	971	0.028	L	0, 38, −27	4.26
					L	−2, 27, −15	4.02

BA = Brodmann area; R = right hemisphere; L = heft hemisphere; MNI = Montreal Neurological Institute; FDR = false discovery rate.

**Table 3 biology-12-00752-t003:** Clusters of negative changes in gray matter (GM) and white matter (WM) associated with brain-PAD revealed in the CUD and matched HC groups, respectively, through multiple regression analysis using SPM12 software.

Analysis	Cluster	Region	BA	Cluster Size (No. of Voxels)	p (PWE)	Hemisphere	MNI Coordinates (x, y, z)	T Value (Peak Voxel)
GM in CUD	1	Frontal Lobe/Limbic Lobe/Subcallosal Gyrus/Parahippocampal Gyrus	34/47	327	0.000	L	−24, 6, −15	6.79
	2	Limbic Lobe/Cingulate Gyrus	24/32	150	0.001	R	8, −3, 45	6.06
						R	4, 6, 46	5.69
	3	Limbic Lobe/Cingulate Gyrus/Anterior Cingulate	32/9	239	0.000	R	4, 28, 33	6.04
						R	8, 18, 40	5.95
	4	Frontal Lobe/Middle Frontal Gyrus/Limbic Lobe/Cingulate Gyrus	6/8/32	253	0.000	L	−2, 32, 40	6.02
						L	−10, 38, 40	5.70
						L	−12, 26, 46	5.57
WM in HC	1	Frontal Lobe/Subgyral	-	57	0.007	L	42, 24, 18	5.95

BA = Brodmann area; R = right hemisphere; L = heft hemisphere; MNI = Montreal Neurological Institute; FDR = false discovery rate.

## Data Availability

The data used in this article were obtained from the SUDMEX CONN (https://openneuro.org/datasets/ds003346/, accessed on 10 December 2022), the Open Access Series of Imaging Studies (OASIS) (https://www.oasis-brains.org//, accessed on 10 December 2022), and the IXI (http://brain-development.org/ixi-dataset//, accessed on 10 December 2022) datasets.

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
