# Peer review of "Cocaine Destroys Gray Matter Brain Cells and Accelerates Brain Aging"

_biology, 2023, doi:10.3390/biology12050752_

Round 1

Reviewer 1 Report

This research analyzes the effects of cocaine on the brain structure and aging process in individuals with cocaine use disorder (CUD). I have identified the following research gaps and limitations of this paper:

1. Introduction:

a) The introduction starts by describing the effects of cocaine on the brain without providing any background information on the prevalence of cocaine use, its impact on society, or why it is essential to study its effects. b) Although the introduction mentions several studies that have investigated the effects of cocaine on the brain, it does not provide a comprehensive literature review. 

2. Materials and methods:

a) The study used a specific imaging protocol and MRI scanner, which may limit the generalizability of the findings to other imaging protocols or scanners. Adding more information on this might be helpful.

b) The author uses a support vector regression algorithm to predict brain age, but details about the support vector regression algorithm are missing. Also, other machine learning or deep learning approaches can be tested and reported in the paper.

3. Results: The study recruited participants from a specific dataset of patients with CUD and HCs from Mexico. Also, the study's sample size is relatively small, with 74 CUD participants and 64 HCs. Therefore, the study's results may not be generalizable to other populations.

Author Response

We would like to thank all reviewers for providing valuable feedback to us. We have adopted suggestions and reflected them in the revised manuscript. Please refer to the highlighted sections (in green) in the manuscript for a detailed revision. In addition, supporting responses to the reviewer's comments are given in bold black.

Responses to comments from Reviewer #1

This research analyzes the effects of cocaine on the brain structure and aging process in individuals with cocaine use disorder (CUD). I have identified the following research gaps and limitations of this paper:

·      I would like to thank the reviewer’s insightful comments. I deeply appreciate the time and efforts you spent on our manuscript. 

  1. Introduction:
  2. a) The introduction starts by describing the effects of cocaine on the brain without providing any background information on the prevalence of cocaine use, its impact on society, or why it is essential to study its effects. b) Although the introduction mentions several studies that have investigated the effects of cocaine on the brain, it does not provide a comprehensive literature review. 
  • The reviewer is quite correct; the effects of cocaine on society should have been mentioned. The following passage has been added in the revised version:

“Cocaine, a potent stimulant drug, is primarily sourced from the coca plant that is grown in South America. Dating back to ancient times, it has a lengthy history of use for both medicinal and recreational purposes. Nevertheless, the widespread consumption of cocaine has become a significant matter of concern for public health due to its addictive attributes and the profound adverse consequences it inflicts on individuals, families, and communities. It has been estimated that 18 million people worldwide are currently using cocaine, with the majority of users located in Latin America, North America, and Europe, according to the World Drug Report 2021 published by the United Nations Office on Drugs and Crime (UNODC; https://www.unodc.org/unodc/data-and-analysis/wdr2021.html). Cocaine usage has a profound and broad impact on society. Cocaine users are at a heightened risk of encountering a wide range of health issues, including cardiovascular disease, respiratory failure, seizures, and mental health disorders like anxiety and depression. In addition, the economic ramifications of cocaine consumption are considerable, as it results in elevated healthcare costs, criminal behavior, and lost productivity.”

  • In the revised version, more papers related to the impact of cocaine on the brain have been added and discussed.

  1. Materials and methods:
  2. a) The study used a specific imaging protocol and MRI scanner, which may limit the generalizability of the findings to other imaging protocols or scanners. Adding more information on this might be helpful.

  • I agree with the reviewer. The following passage has been added to the limitations of this study:

“In this study, we used a specific imaging protocol and MRI scanner, which may limit the generalizability of the findings to other imaging protocols or scanners. Consequently, it is imperative to conduct further studies with larger sample sizes obtained from different MRI protocols to authenticate our findings.”

  1. b) The author uses a support vector regression algorithm to predict brain age, but details about the support vector regression algorithm are missing. Also, other machine learning or deep learning approaches can be tested and reported in the paper.

  • The details of a Support Vector Regression algorithm used in this study have been added in the revised version.
  • Please note that the focus of this study was on the clinical aspects of CUD on brain health. Therefore, we used a standard SVR algorithm, which has been widely used in neuroimaging-driven studies to estimate brain age [1]. I confirm that other machine learning or deep learning approaches may generate different brain age values [2]. The following passage has been added to the discussion part:

“In this study, we employed a standard support vector regression (SVR) algorithm for predicting brain age values, as this approach is commonly used in neuroimaging studies for estimating brain age  [1]. However, it is crucial to acknowledge that several factors, such as the choice of prediction algorithm, image preprocessing, MRI protocols, and bias adjustments, can influence brain age values and affect the interpretation of the results. “

  1. Results:The study recruited participants from a specific dataset of patients with CUD and HCs from Mexico. Also, the study's sample size is relatively small, with 74 CUD participants and 64 HCs. Therefore, the study's results may not be generalizable to other populations.
  • This is an interesting point that the reviewer raises, and they are correct that race or ethnicity of cocaine users can potentially affect their behavioral tendencies. I have added the following passage to the Discussion:

“The sample size used in this study for brain age analysis was limited, consisting of only 74 CUD participants and 64 healthy controls. To gain further insight into the effects of cocaine on brain structure, larger sample sizes should be utilized in future research. Additionally, it is important to acknowledge that this study utilized a specific dataset of CUD patients and HCs from Mexico [3]. The race or ethnicity of individuals who use cocaine may have an impact on their behavioral tendencies, and future studies should aim to investigate the effects of cocaine on brain structure in different racial and ethnic groups.”

References:

[1]        S. Mishra, I. Beheshti, and P. Khanna, "A Review of Neuroimaging-driven Brain Age Estimation for identification of Brain Disorders and Health Conditions," (in eng), IEEE Rev Biomed Eng, vol. PP, Aug 24 2021, doi: 10.1109/RBME.2021.3107372.

[2]        I. Beheshti, M. A. Ganaie, V. Paliwal, A. Rastogi, I. Razzak, and M. Tanveer, "Predicting Brain Age Using Machine Learning Algorithms: A Comprehensive Evaluation," (in eng), IEEE J Biomed Health Inform, vol. 26, no. 4, pp. 1432-1440, 04 2022, doi: 10.1109/JBHI.2021.3083187.

[3]        D. Angeles-Valdez et al., "The Mexican magnetic resonance imaging dataset of patients with cocaine use disorder: SUDMEX CONN," (in eng), Sci Data, vol. 9, no. 1, p. 133, Mar 31 2022, doi: 10.1038/s41597-022-01251-3.

Reviewer 2 Report

The study aims to investigate the effects of cocaine use on global brain health in terms of brain age metric. The authors used a cross-sectional design to compare the brain age of cocaine users and non-users. The study included 120 participants, 60 cocaine users, and 60 non-users, who underwent magnetic resonance imaging (MRI) scans. The authors used a brain-predicted age deviation (brain-PAD) metric to estimate the brain age of the participants. Overall, the study is well-designed and the methodology is appropriate for the research question. The authors used a rigorous statistical analysis to compare the brain age of cocaine users and non-users. The results of the study showed that cocaine use is associated with accelerated brain aging and destruction of gray matter brain cells. The authors also identified specific brain regions that are linked to cocaine use disorder (CUD), which can significantly contribute to our understanding of the impact of cocaine on the brain mechanisms. However, there are some limitations to the study that need to be addressed. Firstly, the study used a cross-sectional design, which limits the ability to establish causality. A longitudinal study design would be more appropriate to investigate the long-term effects of cocaine use on brain aging. Secondly, the study did not control for other factors that may affect brain aging, such as alcohol use, smoking, and other drug use. These factors may confound the results of the study. Thirdly, the study did not investigate the effects of different doses and durations of cocaine use on brain aging. It is possible that the effects of cocaine use on brain aging may vary depending on the dose and duration of use. In conclusion, the study provides important insights into the effects of cocaine use on brain aging and gray matter brain cells. The authors used a brain age metric to quantify the effects of cocaine use on global brain health, which is a novel approach. However, the study has some limitations that need to be addressed in future research. In conclusions, the study contributes to our understanding of the impact of cocaine on the brain mechanisms and can aid in the development of innovative treatments for cocaine use disorder.

I would suggest the following improvements to make it acceptable for publication in a Biology journal: 1. The authors should acknowledge the limitations of the study, such as the cross-sectional design, lack of control for other factors that may affect brain aging, and the need for longitudinal investigations to determine the course of GM atrophy in CUD over time. 2. The authors should provide more information on the study population, such as the duration and frequency of cocaine use, and the severity of CUD. This information would help readers understand the extent of the impact of cocaine on brain aging. 3. The authors should discuss the potential reversibility of brain injuries due to cocaine consumption, for example, by quitting cocaine use and adopting a healthy lifestyle. This discussion would provide readers with a better understanding of the potential for recovery from cocaine-induced brain damage. 4. The authors should provide more information on the brain age metric used in the study, including its validity and reliability. This information would help readers understand the significance of the brain age deviation (brain-PAD) metric in estimating brain age. 5. The authors should include a discussion on the implications of the study for the treatment of CUD and the development of innovative treatments for cocaine-induced brain damage. 6. The authors should provide more information on the statistical analysis used in the study, including the methods used to control for confounding variables and the significance level used to determine statistical significance. 7. The authors should include a discussion on the potential mechanisms underlying the effects of cocaine on brain aging, such as oxidative stress, inflammation, and neurotoxicity. This discussion would provide readers with a better understanding of the biological processes involved in cocaine-induced brain damage. 8. The authors should provide more information on the imaging techniques used in the study, including the resolution and quality of the MRI scans. This information would help readers understand the accuracy and reliability of the imaging data. 9. Include a discussion on the potential confounding effects of weight loss on brain health. 10. The authors should clarify the study's contribution to the existing literature on the effects of cocaine on brain aging and gray matter brain cells. This clarification would help readers understand the novelty and significance of the study's findings. 11. The authors should provide more information on the participants' demographic characteristics, such as their age, gender, and ethnicity. This information would help readers understand the generalizability of the study's findings to other populations. 12. The authors should include a discussion on the potential clinical implications of the study, such as the development of new treatments for CUD and the use of brain age estimation as a biomarker for identifying brain disorders and health conditions.

Author Response

We would like to thank all reviewers for providing valuable feedback to us. We have adopted suggestions and reflected them in the revised manuscript. Please refer to the highlighted sections (in green) in the manuscript for a detailed revision. In addition, supporting responses to the reviewer's comments are given in bold black.

Responses to comments from Reviewer #2

The study aims to investigate the effects of cocaine use on global brain health in terms of brain age metric. The authors used a cross-sectional design to compare the brain age of cocaine users and non-users. The study included 120 participants, 60 cocaine users, and 60 non-users, who underwent magnetic resonance imaging (MRI) scans. The authors used a brain-predicted age deviation (brain-PAD) metric to estimate the brain age of the participants. Overall, the study is well-designed and the methodology is appropriate for the research question. The authors used a rigorous statistical analysis to compare the brain age of cocaine users and non-users. The results of the study showed that cocaine use is associated with accelerated brain aging and destruction of gray matter brain cells. The authors also identified specific brain regions that are linked to cocaine use disorder (CUD), which can significantly contribute to our understanding of the impact of cocaine on the brain mechanisms. However, there are some limitations to the study that need to be addressed. Firstly, the study used a cross-sectional design, which limits the ability to establish causality. A longitudinal study design would be more appropriate to investigate the long-term effects of cocaine use on brain aging. Secondly, the study did not control for other factors that may affect brain aging, such as alcohol use, smoking, and other drug use. These factors may confound the results of the study. Thirdly, the study did not investigate the effects of different doses and durations of cocaine use on brain aging. It is possible that the effects of cocaine use on brain aging may vary depending on the dose and duration of use. In conclusion, the study provides important insights into the effects of cocaine use on brain aging and gray matter brain cells. The authors used a brain age metric to quantify the effects of cocaine use on global brain health, which is a novel approach. However, the study has some limitations that need to be addressed in future research. In conclusions, the study contributes to our understanding of the impact of cocaine on the brain mechanisms and can aid in the development of innovative treatments for cocaine use disorder. I would suggest the following improvements to make it acceptable for publication in a Biology journal:

·      I would like to extend my heartfelt appreciation to the anonymous reviewer for their valuable feedback and constructive comments on our paper. Their insightful suggestions have significantly improved the quality and clarity of my work.

  1. The authors should acknowledge the limitations of the study, such as the cross-sectional design, lack of control for other factors that may affect brain aging, and the need for longitudinal investigations to determine the course of GM atrophy in CUD over time.
  • The reviewer is quite correct that the limitations of this study should be highlighted. As suggested, the limitations of this study have been amended and highlighted in the discussion section.
  1. The authors should provide more information on the study population, such as the duration and frequency of cocaine use, and the severity of CUD. This information would help readers understand the extent of the impact of cocaine on brain aging.
  • More information about participants used in this study has been added into section 2.1 and Table 1.
  1. The authors should discuss the potential reversibility of brain injuries due to cocaine consumption, for example, by quitting cocaine use and adopting a healthy lifestyle. This discussion would provide readers with a better understanding of the potential for recovery from cocaine-induced brain damage.
  • The main objective of this study was to identify and quantify the brain changes caused by cocaine consumption. We demonstrated that cocaine has a negative impact on brain structure, particularly the cortex area. However, it is unclear whether these brain injuries due to cocaine consumption are reversible, for example, by quitting cocaine use and adopting a healthy diet, stress management, and exercise. This point has been highlighted in the discussion section.
  1. The authors should provide more information on the brain age metric used in the study, including its validity and reliability. This information would help readers understand the significance of the brain age deviation (brain-PAD) metric in estimating brain age.
  • This is an excellent suggestion. I have added the following passage to the Discussion:

“Brain age estimation is a technique that utilizes supervised machine learning algorithms to analyze brain scans and determine the biological age of an individual's brain based on their brain features. This technique involves compressing the whole-brain information into a single numerical value, known as the Brain-PAD, which represents the difference between an individual's predicted brain age and their chronological age. A negative Brain-PAD score suggests a younger-looking brain, which is advantageous, while a positive score may indicate a divergence between predicted and chronological age, potentially indicating an increased risk of age-related cognitive decline or disease. Brain age estimation has proven useful in detecting early signs of neurodegenerative diseases, assessing overall brain health and aging, predicting cognitive decline, and identifying the impact of lifestyle factors such as diet, exercise, mental health, and stress on brain aging. Further information about the brain age estimation technique and its potential application in clinical settings can be found in [1, 2].”

  1. The authors should include a discussion on the implications of the study for the treatment of CUD and the development of innovative treatments for cocaine-induced brain damage.

  • Transcranial magnetic stimulation (rTMS) and high-definition transcranial direct current stimulation (HD-tDCS) are promising and safe novel treatment options for individuals with CUD. However, the primary challenge in utilizing these techniques is determining the most precise areas to stimulate. Therefore, one of the primary objectives of this study was to identify the specific brain regions associated with CUD using VBM and Brain-age estimation techniques. These identified areas may represent potential targets for rTMS or HD-tDCS therapies, which could be a significant contribution to the field. This aspect has been emphasized in the discussion section.

  1. The authors should provide more information on the statistical analysis used in the study, including the methods used to control for confounding variables and the significance level used to determine statistical significance.
  • The details of the statistical analysis employed in this study, encompassing techniques, covariates, and levels of significance, have been revised and elaborated upon in Section 2.3.

  1. The authors should include a discussion on the potential mechanisms underlying the effects of cocaine on brain aging, such as oxidative stress, inflammation, and neurotoxicity. This discussion would provide readers with a better understanding of the biological processes involved in cocaine-induced brain damage.

  • That's a great recommendation. I have incorporated the following statement into the introduction section:

“Cocaine can adversely affect the brain through various biological mechanisms, such as oxidative stress, inflammation, and neurotoxicity. Oxidative stress occurs due to an imbalance between the formation of reactive oxygen species (ROS) and the body's ability to eliminate them [3]. Cocaine increases the production of ROS, which in turn leads to oxidative damage to brain cells  [4]. Moreover, cocaine usage can induce an inflammatory response within the brain  [5], resulting in a multitude of negative consequences, including damage to brain cells. Neurotoxicity is another direct consequence of cocaine use [6, 7], as it stimulates the release of neurotransmitters such as dopamine, serotonin, and norepinephrine, which can cause excitotoxicity, leading to harm or demise of brain cells.”

  1. The authors should provide more information on the imaging techniques used in the study, including the resolution and quality of the MRI scans. This information would help readers understand the accuracy and reliability of the imaging data.
  • The details of MRI scans and image processing are provided in Section 2. In addition, I have added the details of image quality assessments in the revised version.

“Besides the visual evaluation, the quality of the images was evaluated using the "Check Homogeneity" feature within the CAT12 toolbox. This function was developed to detect MRI images that may contain artifacts or other forms of variability in voxel intensity values, which could potentially compromise the accuracy and consistency of subsequent analyses.”

  1. Include a discussion on the potential confounding effects of weight loss on brain health.
  • Please note that the objective of this study was to determine the effects of cocaine on brain health. Indeed, substantial weight loss was an example in this study that can reform the brain damage caused by obesity. The details of potential confounding effects of weight loss on brain health can be found in [8].
  1. The authors should clarify the study's contribution to the existing literature on the effects of cocaine on brain aging and gray matter brain cells. This clarification would help readers understand the novelty and significance of the study's findings.

  • The Discussion section of this study provides a thorough analysis of the similarities and differences between our findings and the existing literature concerning VBM analysis. I also added more literature in the revised version. to the best of our knowledge, this study is the first to investigate the impact of cocaine use on global brain health using a brain age estimation technique. These points have been elaborated upon in the discussion portion of this paper.

  1. The authors should provide more information on the participants' demographic characteristics, such as their age, gender, and ethnicity. This information would help readers understand the generalizability of the study's findings to other populations.
  • As recommended, additional information regarding the demographic characteristics of the study participants, including their age, gender, and ethnicity, has been included in Section 2.1.
  1. The authors should include a discussion on the potential clinical implications of the study, such as the development of new treatments for CUD and the use of brain age estimation as a biomarker for identifying brain disorders and health conditions.

  • The primary objective of this study was to investigate the impact of cocaine use on brain function and health through the utilization of statistical analysis (VBM) and machine learning models (Brain Age Estimation Technique). Moreover, we aimed to identify specific brain regions associated with CUD that could potentially serve as suitable targets for rTMS or HD-tDCS therapies. The discussion section of this paper elaborates on these key findings.

References:

[1]        S. Mishra, I. Beheshti, and P. Khanna, "A Review of Neuroimaging-driven Brain Age Estimation for identification of Brain Disorders and Health Conditions," IEEE Reviews in Biomedical Engineering, 2021.

[2]        D. Sone and I. Beheshti, "Neuroimaging-based brain age estimation: a promising personalized biomarker in neuropsychiatry," Journal of Personalized Medicine, vol. 12, no. 11, p. 1850, 2022.

[3]        D. Cerretani, V. Fineschi, S. Bello, I. Riezzo, E. Turillazzi, and M. Neri, "Role of oxidative stress in cocaine-induced cardiotoxicity and cocaine-related death," (in eng), Curr Med Chem, vol. 19, no. 33, pp. 5619-23, 2012, doi: 10.2174/092986712803988785.

[4]        J.-B. Dietrich, A. Mangeol, M.-O. Revel, C. Burgun, D. Aunis, and J. Zwiller, "Acute or repeated cocaine administration generates reactive oxygen species and induces antioxidant enzyme activity in dopaminergic rat brain structures," Neuropharmacology, vol. 48, no. 7, pp. 965-974, 2005.

[5]        C. Correia, P. Romieu, M. C. Olmstead, and K. Befort, "Can cocaine-induced neuroinflammation explain maladaptive cocaine-associated memories?," (in eng), Neurosci Biobehav Rev, vol. 111, pp. 69-83, Apr 2020, doi: 10.1016/j.neubiorev.2020.01.001.

[6]        H. S. Sharma, D. Muresanu, A. Sharma, and R. Patnaik, "Cocaine-induced breakdown of the blood–brain barrier and neurotoxicity," International review of neurobiology, vol. 88, pp. 297-334, 2009.

[7]        R. B. Pereira, P. B. Andrade, and P. Valentão, "A Comprehensive View of the Neurotoxicity Mechanisms of Cocaine and Ethanol," (in eng), Neurotox Res, vol. 28, no. 3, pp. 253-67, Oct 2015, doi: 10.1007/s12640-015-9536-x.

[8]        Y. Zeighami et al., "Impact of weight loss on brain age: Improved brain health following bariatric surgery," (in eng), Neuroimage, vol. 259, p. 119415, Oct 01 2022, doi: 10.1016/j.neuroimage.2022.119415.

  •  

Reviewer 3 Report

This paper is interesting, but some points need to be revised:

- "At the second level, we used the brain age.." Does the author mean a secondary goal?

- "To date, the brain age estimation paradigm has been widely used to assess the impact of different neurological diseases – e.g., Alzheimer’s disease [15], epilepsy [16], Parkinson's disease [17] and schizophrenia [18] –  on brain health status" but also stroke. Look at Chavda et al. Ischemic Stroke and.. DOI: 10.3390/neurolint14020032

- What does author suggest about this ?"The identification of the particular brain regions affected by CUD could greatly enhance our comprehension of the disorder's effects on brain structure and potentially lead to the development of novel therapeutic approaches for individuals with this condition"

- "Consequently, these findings support our hypothesis that individuals with CUD experience considerably faster brain aging. A mean brain-PAD of +2.5 years in CUD patients is similar to that of some neurological diseases, such as Parkinson's disease (+2.5 years)[17], schizophrenia (+2.56 years) ... " Look at these important and recent refs. --  doi: 10.3390/ijerph19148799  -- doi: 10.3390/jcm11195844

- "Damage to the cingulate gyrus might impair the ability to respond to some stimuli, leading to aggressive behavior, decreased emotional expressiveness, or shyness." Improve this point.

- "Furthermore, additional research should be conducted to investigate the impact of cocaine on functional brain deficits, such as functional connectivity.Which kind of research paper does the author suggest? doi: 10.1038/s41598-020-76182-3

- Figure 3 legend should be improved.

Quite good

Author Response

We would like to thank all reviewers for providing valuable feedback to us. We have adopted suggestions and reflected them in the revised manuscript. Please refer to the highlighted sections (in green) in the manuscript for a detailed revision. In addition, supporting responses to the reviewer's comments are given in bold black.

Responses to comments from Reviewer #3

This paper is interesting, but some points need to be revised:

·      I would like to express my sincere gratitude to the reviewer for taking the time to thoroughly review my manuscript and provide valuable feedback. Their insightful comments and suggestions have greatly improved the quality and clarity of the article. I appreciate the reviewer's efforts in enhancing the scientific rigor of our study. Thank you again for your time and expertise.

- "At the second level, we used the brain age.." Does the author mean a secondary goal?

  • As stated in the abstract and main text, the VBM technique was employed at the first level of analysis to identify morphological changes in individuals with cocaine use disorder (CUD) compared to healthy controls (HCs). Furthermore, the impact of cocaine on global brain health was assessed at the second level through the use of brain age estimation techniques.

- "To date, the brain age estimation paradigm has been widely used to assess the impact of different neurological diseases – e.g., Alzheimer’s disease [15], epilepsy [16], Parkinson's disease [17] and schizophrenia [18] –  on brain health status" but also stroke. Look at Chavda et al. Ischemic Stroke and.. DOI: 10.3390/neurolint14020032

  • I appreciate your suggestion to include the reference “DOI: 10.3390/neurolint14020032” in the manuscript. After carefully reviewing this reference, I have found that it is not directly related to assessing brain health through the brain age estimation technique. Please note that main purpose of the above sentences ("To date, the brain age estimation paradigm has been widely used to assess the impact of different neurological diseases – e.g., Alzheimer’s disease [15], epilepsy [16], Parkinson's disease [17] and schizophrenia [18] –  on brain health status") was to introduce the application of brain age estimation technique in different brain diseases.

- What does author suggest about this ?"The identification of the particular brain regions affected by CUD could greatly enhance our comprehension of the disorder's effects on brain structure and potentially lead to the development of novel therapeutic approaches for individuals with this condition"

  • Identifying brain regions associated with CUD can offer significant insights into the underlying mechanisms of this disease. A better understanding of the pathophysiology of CUD can facilitate the development of new treatments and therapies. Our study found widespread gray matter atrophy in CUD patients, located in the temporal lobe, frontal lobe, insula, middle frontal gyrus, superior frontal gyrus, rectal gyrus, and limbic lobe regions, compared to HCs. By investigating the functional role of these brain regions in the disease progression, as well as the molecular and cellular mechanisms disrupted in these regions, we may gain insight into how CUD develops. Additionally, identifying brain regions associated with CUD can enable researchers to develop more targeted treatments, such as rTMS or HD-tDCS therapies, for CUD. We have emphasized this point in the discussion section of our manuscript.

- "Consequently, these findings support our hypothesis that individuals with CUD experience considerably faster brain aging. A mean brain-PAD of +2.5 years in CUD patients is similar to that of some neurological diseases, such as Parkinson's disease (+2.5 years)[17], schizophrenia (+2.56 years) ... " Look at these important and recent refs. --  doi: 10.3390/ijerph19148799  -- doi: 10.3390/jcm11195844

  • Regarding your suggestion to include references “doi: 10.3390/ijerph19148799” and “doi: 10.3390/jcm11195844” in my paper, I have carefully considered this recommendation. After reviewing these references, I have concluded that they are not directly relevant to the focus of my research. While I understand that these references may be of interest to readers who want to explore related topics, I believe that including them would not add significant value to my paper and might distract from the main argument. If you have any further suggestions for improving my paper, I would be happy to consider them. Please note that the purpose of the above passage was to compare the mean Brain-PAD in cocaine with some neurological diseases that showed similar brain age deviations.

- "Damage to the cingulate gyrus might impair the ability to respond to some stimuli, leading to aggressive behavior, decreased emotional expressiveness, or shyness." Improve this point.

Response:

  • As suggested, the following passage has been added into the discussion section:

“Damage to the cingulate gyrus might impair the ability to respond to some stimuli, leading to aggressive behavior, decreased emotional expressiveness, or shyness. Indeed, damage to the cingulate gyrus has been linked with increased aggression and impulsivity, as well as decreased empathy and social awareness [1]. These changes may result from an inability to regulate emotional responses or to effectively interpret social cues, which can lead to difficulties in social interactions. Thus, damage to the cingulate gyrus can have significant negative impacts on a person's emotional and behavioral functioning, underscoring the significance of this brain region in the regulation and processing of emotions [2].”

- "Furthermore, additional research should be conducted to investigate the impact of cocaine on functional brain deficits, such as functional connectivity." Which kind of research paper does the author suggest? doi: 10.1038/s41598-020-76182-3

  • Functional connectivity is important to study because it helps us understand the organization and communication of neural networks in the brain and functional interactions among brain regions. Abnormal patterns of functional connectivity have been documented in different neurological diseases, such as AD and PD, that may reflect disruptions in the communication between different brain regions. The following passage has been added to the revised version.

“Furthermore, additional research should be conducted to investigate the impact of cocaine on functional brain deficits, such as functional connectivity. Particularly, investigating functional connectivity through resting-state fMRI data is essential to comprehending the arrangement and communication of neural networks in the brain, as well as the functional interactions between different brain regions [3]”.

- Figure 3 legend should be improved.

  • 3 has been modified.

References:

[1]        O. Devinsky, M. J. Morrell, and B. A. Vogt, "Contributions of anterior cingulate cortex to behaviour," Brain, vol. 118, no. 1, pp. 279-306, 1995.

[2]        B. A. Vogt, "Pain and emotion interactions in subregions of the cingulate gyrus," Nature Reviews Neuroscience, vol. 6, no. 7, pp. 533-544, 2005.

[3]        K. Xu, Y. Liu, Y. Zhan, J. Ren, and T. Jiang, "BRANT: a versatile and extendable resting-state fMRI toolkit," Frontiers in neuroinformatics, vol. 12, p. 52, 2018.

Round 2

Reviewer 1 Report

Thanks for solving and discussing the previous comments. Now I am satisfied with the present version of the paper.

Reviewer 2 Report

Well revised. The manuscript can be accepted for publication.

Reviewer 3 Report

Good